# DropBlock: A regularization method for convolutional networks

**Golnaz Ghiasi**
Google Brain

**Tsung-Yi Lin**
Google Brain

**Quoc V. Le**
Google Brain

## Abstract

Deep neural networks often work well when they are over-parameterized and trained with a massive amount of noise and regularization, such as weight decay and dropout. Although dropout is widely used as a regularization technique for fully connected layers, it is often less effective for convolutional layers. This lack of success of dropout for convolutional layers is perhaps due to the fact that activation units in convolutional layers are spatially correlated so information can still flow through convolutional networks despite dropout. Thus a structured form of dropout is needed to regularize convolutional networks. In this paper, we introduce DropBlock, a form of structured dropout, where units in a contiguous region of a feature map are dropped together. We found that applying DropbBlock in skip connections in addition to the convolution layers increases the accuracy. Also, gradually increasing number of dropped units during training leads to better accuracy and more robust to hyperparameter choices. Extensive experiments show that DropBlock works better than dropout in regularizing convolutional networks. On ImageNet classification, ResNet-50 architecture with DropBlock achieves 78.13% accuracy, which is more than 1.6% improvement on the baseline. On COCO detection, DropBlock improves Average Precision of RetinaNet from 36.8% to 38.4%.

## 1   Introduction

Deep neural nets work well when they have a large number of parameters and are trained with a massive amount of regularization and noise, such as weight decay and dropout [1]. Though the first biggest success of dropout was associated with convolutional networks [2], recent convolutional architectures rarely use dropout [3, 4, 5, 6, 7, 8, 9, 10]. In most cases, dropout was mainly used at the fully connected layers of the convolutional networks [11, 12, 13].

We argue that the main drawback of dropout is that it drops out features randomly. While this can be effective for fully connected layers, it is less effective for convolutional layers, where features are correlated spatially. When the features are correlated, even with dropout, information about the input can still be sent to the next layer, which causes the networks to overfit. This intuition suggests that a more structured form of dropout is needed to better regularize convolutional networks.

In this paper, we introduce DropBlock, a structured form of dropout, that is particularly effective to regularize convolutional networks. In DropBlock, features in a block, i.e., a contiguous region of a feature map, are dropped together. As DropBlock discards features in a correlated area, the networks must look elsewhere for evidence to fit the data (see Figure 1).

In our experiments, DropBlock is much better than dropout in a range of models and datasets. Adding DropBlock to ResNet-50 architecture improves image classification accuracy on ImageNet from 76.51% to 78.13%. On COCO detection, DropBlock improves AP of RetinaNet from 36.8% to 38.4%.

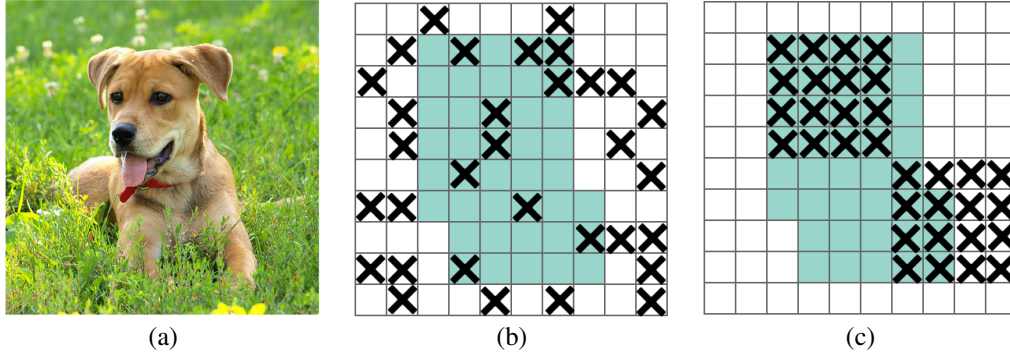

(a)           (b)           (c)

Figure 1: (a) input image to a convolutional neural network. The green regions in (b) and (c) include the activation units which contain semantic information in the input image. Dropping out activations at random is not effective in removing semantic information because nearby activations contain closely related information. Instead, dropping continuous regions can remove certain semantic information (e.g., head or feet) and consequently enforcing remaining units to learn features for classifying input image.

## 2 Related work

Since its introduction, dropout [1] has inspired a number of regularization methods for neural networks such as DropConnect [14], maxout [15], StochasticDepth [16], DropPath [17], Sched-uledDropPath [8], shake-shake regularization [18], and ShakeDrop regularization [19]. The basic principle behind these methods is to inject noise into neural networks so that they do not overfit the training data. When it comes to convolutional neural networks, most successful methods require the noise to be structured [16, 17, 8, 18, 19, 20]. For example, in DropPath, an entire layer in the neural network is zeroed out of training, not just a particular unit. Although these strategies of dropping out layers may work well for layers with many input or output branches, they cannot be used for layers without any branches. Our method, DropBlock, is more general in that it can be applied anywhere in a convolutional network. Our method is closely related to SpatialDropout [20], where an entire channel is dropped from a feature map. Our experiments show that DropBlock is more effective than SpatialDropout.

The developments of these noise injection techniques specific to the architectures are not unique to convolutional networks. In fact, similar to convolutional networks, recurrent networks require their own noise injection methods. Currently, Variational Dropout [21] and ZoneOut [22] are two of the most commonly used methods to inject noise to recurrent connections.

Our method is inspired by Cutout [23], a data augmentation method where parts of the input examples are zeroed out. DropBlock generalizes Cutout by applying Cutout at every feature map in a convolutional networks. In our experiments, having a fixed zero-out ratio for DropBlock during training is not as robust as having an increasing schedule for the ratio during training. In other words, it's better to set the DropBlock ratio to be small initially during training, and linearly increase it over time during training. This scheduling scheme is related to ScheduledDropPath [8].

## 3 DropBlock

DropBlock is a simple method similar to dropout. Its main difference from dropout is that it drops contiguous regions from a feature map of a layer instead of dropping out independent random units. Pseudocode of DropBlock is shown in Algorithm 1. DropBlock has two main parameters which are $block\_size$ and $\gamma$. $block\_size$ is the size of the block to be dropped, and $\gamma$, controls how many activation units to drop.

We experimented with a shared DropBlock mask across different feature channels or each feature channel has its DropBlock mask. Algorithm 1 corresponds to the latter, which tends to work better in our experiments.

---

**Algorithm 1** DropBlock

---

1: **Input:**output activations of a layer ($A$), $block\_size$, $\gamma$, $mode$
2: **if** $mode == Inference$ **then**
3:     return $A$
4: **end if**
5: Randomly sample mask $M$: $M_{i,j} \sim Bernoulli(\gamma)$
6: For each zero position $M_{i,j}$, create a spatial square mask with the center being $M_{i,j}$, the width, height being $block\_size$ and set all the values of $M$ in the square to be zero (see Figure 2).
7: Apply the mask: $A = A \times M$
8: Normalize the features: $A = A \times \textbf{count}(M)/\textbf{count\_ones}(M)$

---

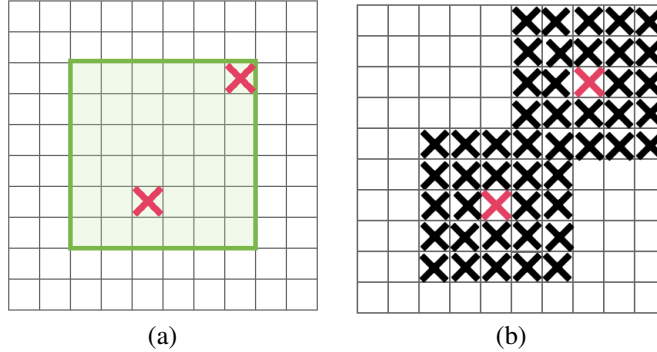

(a)                    (b)

Figure 2: Mask sampling in DropBlock. (a) On every feature map, similar to dropout, we first sample a mask $M$. We only sample mask from shaded green region in which each sampled entry can expanded to a mask fully contained inside the feature map. (b) Every zero entry on $M$ is expanded to $block\_size \times block\_size$ zero block.

Similar to dropout we do not apply DropBlock during inference. This is interpreted as evaluating an averaged prediction across the exponentially-sized ensemble of sub-networks. These sub-networks include a special subset of sub-networks covered by dropout where each network does not see contiguous parts of feature maps.

**Setting the value of** $block\_size$**.**   In our implementation, we set a constant $block\_size$ for all feature maps, regardless the resolution of feature map. DropBlock resembles dropout [1] when $block\_size = 1$ and resembles SpatialDropout [20] when $block\_size$ covers the full feature map.

**Setting the value of** $\gamma$**.**   In practice, we do not explicitly set $\gamma$. As stated earlier, $\gamma$ controls the number of features to drop. Suppose that we want to keep every activation unit with the probability of $keep\_prob$, in dropout [1] the binary mask will be sampled with the Bernoulli distribution with mean $1 - keep\_prob$. However, to account for the fact that every zero entry in the mask will be expanded by $block\_size^2$ and the blocks will be fully contained in feature map, we need to adjust $\gamma$ accordingly when we sample the initial binary mask. In our implementation, $\gamma$ can be computed as

$$\gamma = \frac{1 - keep\_prob}{block\_size^2} \frac{feat\_size^2}{(feat\_size - block\_size + 1)^2} \tag{1}$$

where $keep\_prob$ can be interpreted as the probability of keeping a unit in traditional dropout. The size of valid seed region is $(feat\_size - block\_size + 1)^2$ where $feat\_size$ is the size of feature map. The main nuance of DropBlock is that there will be some overlapped in the dropped blocks, so the above equation is only an approximation. In our experiments, we first estimate the $keep\_prob$ to use (between 0.75 and 0.95), and then compute $\gamma$ according to the above equation.

**Scheduled DropBlock.**   We found that DropBlock with a fixed $keep\_prob$ during training does not work well. Applying small value of $keep\_prob$ hurts learning at the beginning. Instead, gradually decreasing $keep\_prob$ over time from 1 to the target value is more robust and adds improvement for

the most values of $keep\_prob$. In our experiments, we use a linear scheme of decreasing the value of $keep\_prob$, which tends to work well across many hyperparameter settings. This linear scheme is similar to ScheduledDropPath [8].

# 4 Experiments

In the following sections, we empirically investigate the effectiveness of DropBlock for image classification, object detection, and semantic segmentation. We apply DropBlock to ResNet-50 [4] with extensive experiments for image classification. To verify the results can be transferred to a different architecture, we perform DropBlock on a state-of-the-art model architecture, AmoebaNet [10], and show improvements. In addition to image classification, We show DropBlock is helpful in training RetinaNet [24] for object detection and semantic segmentation.

## 4.1 ImageNet Classification

The ILSVRC 2012 classification dataset [25] contains 1.2 million training images, 50,000 validation images, and 150,000 testing images. Images are labeled with 1,000 categories. We used horizontal flip, scale, and aspect ratio augmentation for training images as in [12, 26]. During evaluation, we applied a single-crop rather than averaging results over multiple crops. Following the common practice, we report classification accuracy on the validation set.

**Implementation Details**   We trained models on Tensor Processing Units (TPUs) and used the official Tensorflow implementations for ResNet-50[1] and AmoebaNet[2]. We used the *default* image size ($224 \times 224$ for ResNet-50 and $331 \times 331$ for AmoebaNet), batch size (1024 for ResNet-50 and 2048 for AmoebaNet) and hyperparameters setting for all the models. We only increased number of training epochs from 90 to 300 for ResNet-50 architecture. The learning rate was decayed by the factor of 0.1 at 100, 200 and 265 epochs. AmoebaNet models were trained for 340 epochs and exponential decay scheme was used for scheduling learning rate. Since baselines are usually overfitted for the longer training scheme and have lower validation accuracy at the end of training, we report the highest validation accuracy over the full training course for fair comparison.

### 4.1.1 DropBlock in ResNet-50

ResNet-50 [4] is a widely used Convolutional Neural Network (CNN) architecture for image recognition. In the following experiments, we apply different regularization techniques on ResNet-50 and compare the results with DropBlock. The results are summarized in Table 1.

| Model | top-1(%) | top-5(%) |
|---|---|---|
| ResNet-50 | $76.51 \pm 0.07$ | $93.20 \pm 0.05$ |
| ResNet-50 + dropout (kp=0.7) [1] | $76.80 \pm 0.04$ | $93.41 \pm 0.04$ |
| ResNet-50 + DropPath (kp=0.9) [17] | $77.10 \pm 0.08$ | $93.50 \pm 0.05$ |
| ResNet-50 + SpatialDropout (kp=0.9) [20] | $77.41 \pm 0.04$ | $93.74 \pm 0.02$ |
| ResNet-50 + Cutout [23] | $76.52 \pm 0.07$ | $93.21 \pm 0.04$ |
| ResNet-50 + AutoAugment [27] | 77.63 | 93.82 |
| ResNet-50 + label smoothing (0.1) [28] | $77.17 \pm 0.05$ | $93.45 \pm 0.03$ |
| ResNet-50 + DropBlock, (kp=0.9) | $78.13 \pm 0.05$ | $94.02 \pm 0.02$ |
| ResNet-50 + DropBlock (kp=0.9) + label smoothing (0.1) | $78.35 \pm 0.05$ | $94.15 \pm 0.03$ |

Table 1: Summary of validation accuracy on ImageNet dataset for ResNet-50 architecture. For dropout, DropPath, and SpatialDropout, we trained models with different $keep\_prob$ values and reported the best result. DropBlock is applied with $block\_size = 7$. We report average over 3 runs. The code of these results is in `https://github.com/tensorflow/tpu/tree/master/models/official/resnet`.

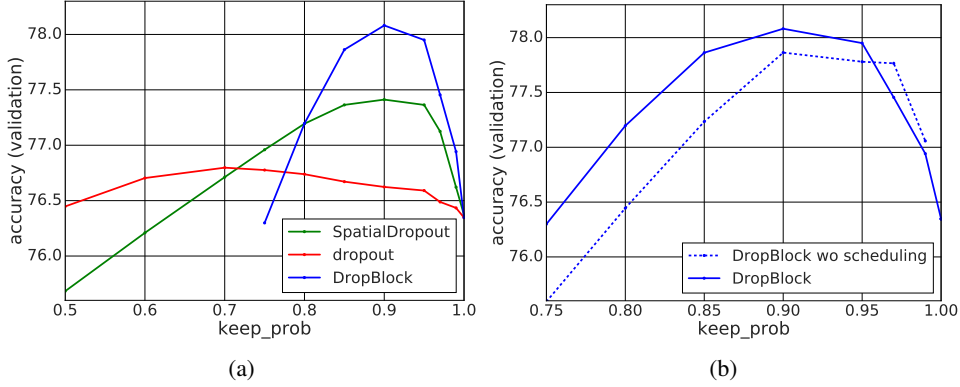

(a)            (b)

Figure 3: ImageNet validation accuracy against $keep\_prob$ with ResNet-50 model. All methods drop activation units in group 3 and 4.

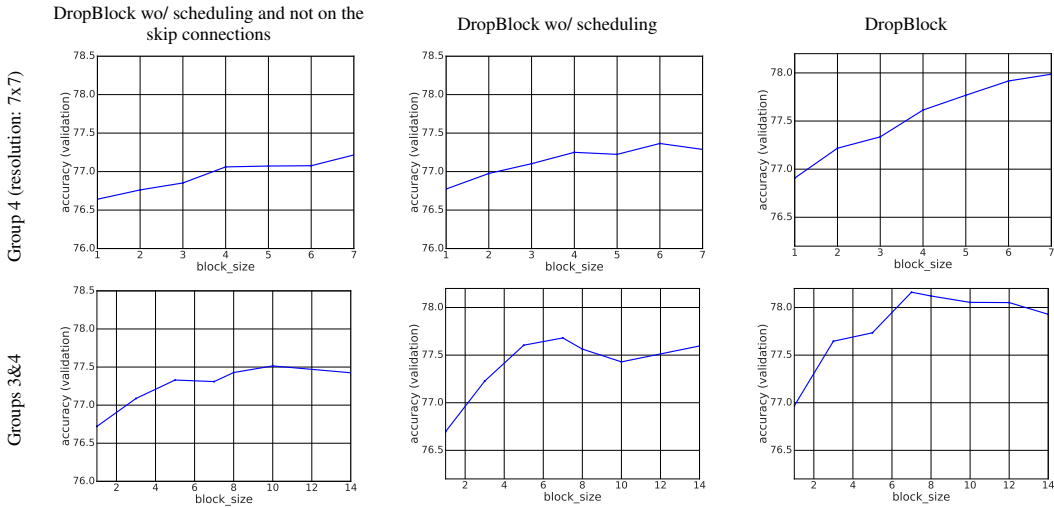

Figure 4: Comparison of ResNet-50 trained on ImageNet when DropBlock is applied to group 4 or groups 3 and 4. From left to right, we show performance of applying DropBlock on convolution branches only and progressively improve it by applying DropBlock on skip connections and adding scheduling $keep\_prob$. The best accuracy is achieved by using $block\_size = 7$ (bottom right figure).

**Where to apply DropBlock.** In residual networks, a building block consists of a few convolution layers and a separate skip connection that performs identity mapping. Every convolution layer is followed by batch normalization layer and ReLU activation. The output of a building block is the sum of outputs from the convolution branch and skip connection.

A residual network can be represented by building groups based on the spatial resolution of feature activation. A building group consists of multiple building blocks. We use group 4 to represent the last group in residual network (i.e., all layers in conv5_x) and so on.

In the following experiments, we study where to apply DropBlock in residual networks. We experimented with applying DropBlock only after convolution layers or applying DropBlock after both convolution layers and skip connections. To study the performance of DropBlock applying to different feature groups, we experimented with applying DropBlock to Group 4 or to both Groups 3 and 4.

**DropBlock vs. dropout.** The original ResNet architecture does not apply any dropout in the model. For the ease of discussion, we define the dropout baseline for ResNet as applying dropout on convolution branches only. We applied DropBlock to both groups 3 and 4 with $block\_size = 7$ by default. We decreased $\gamma$ by factor 4 for group 3 in all the experiments. In Figure 3-(a), we show

that DropBlock outperforms dropout with 1.3% for top-1 accuracy. The scheduled $keep\_prob$ makes DropBlock more robust to the change of $keep\_prob$ and adds improvement for the most values of $keep\_prob$ (3-(b)).

With the best $keep\_prob$ found in Figure 3, we swept over $block\_size$ from 1 to size covering full feature map. Figure 4 shows applying larger $block\_size$ is generally better than applying $block\_size$ of 1. The best DropBlock configuration is to apply $block\_size = 7$ to both groups 3 and 4.

In all configurations, DropBlock and dropout share the similar trend and DropBlock has a large gain compared to the best dropout result. This shows evidence that the DropBlock is a more effective regularizer compared to dropout.

**DropBlock vs. SpatialDropout.**   Similar as dropout baseline, we define the SpatialDropout [20] baseline as applying it on convolution branches only. SpatialDropout is better than dropout but inferior to DropBlock. In Figure 4, we found SpatialDropout can be too harsh when applying to high resolution feature map on group 3. DropBlock achieves the best result by dropping block with constant size on both groups 3 and 4.

**Comparison with DropPath.**   Following ScheduledDropPath [8] we applied scheduled DropPath on all connections except the skip connections. We trained models with different values for $keep\_prob$ parameter. Also, we trained models where we applied DropPath in all groups and similar to our other experiments only at group 4 or at group 3 and 4. We achieved best validation accuracy of 77.10% when we only apply it to group 4 with $keep\_prob = 0.9$.

**Comparison with Cutout.**   We also compared with Cutout [23] which is a data augmentation method and randomly drops a fixed size block from the input images. Although Cutout improves accuracy on the CIFAR-10 dataset as suggested by [23], it does not improve the accuracy on the ImageNet dataset in our experiments.

**Comparison with other regularization techniques.**   We compare DropBlock to data augmentation and label smoothing, which are two commonly used regularization techniques. In Table 1, DropBlock has better performance compared to strong data augmentation [27] and label smoothing [28]. The performance improves when combining DropBlock and label smoothing and train for 330 epochs, showing the regularization techniques can be complimentary when we train for longer.

### 4.1.2   DropBlock in AmoebaNet

We also show the effectiveness of DropBlock on a recent AmoebaNet-B architecture which is a state-of-art architecture, found using evolutionary architecture search [10]. This model has dropout with keep probability of 0.5 but only on the the final softmax layer.

We apply DropBlock after all batch normalization layers and also in the skip connections of the last 50% of the cells. The resolution of the feature maps in these cells are 21x21 or 11x11 for input image with the size of 331x331. Based on the experiments in the last section, we used $keep\_prob$ of 0.9 and set $block\_size = 11$ which is the width of the last feature map. DropBlock improves top-1 accuracy of AmoebaNet-B from 82.25% to 82.52% (Table 2).

| Model | top-1(%) | top-5(%) |
|---|---|---|
| AmoebaNet-B (6, 256) | 82.25 | 95.88 |
| AmoebaNet-B (6, 256) + DropBlock | 82.52 | 96.07 |

Table 2: Top-1 and top-5 validation accuracy of AmoebaNet-B architecture trained on ImageNet.

## 4.2   Experimental Analysis

DropBlock demonstrates strong empirical results on improving ImageNet classification accuracy compared to dropout. We hypothesize dropout is insufficient because the contiguous regions in convolution layers are strongly correlated. Randomly dropping a unit still allows information to flow through neighboring units. In this section, we conduct an analysis to show DropBlock is more effective in dropping semantic information. Subsequently, the model regularized by DropBlock is

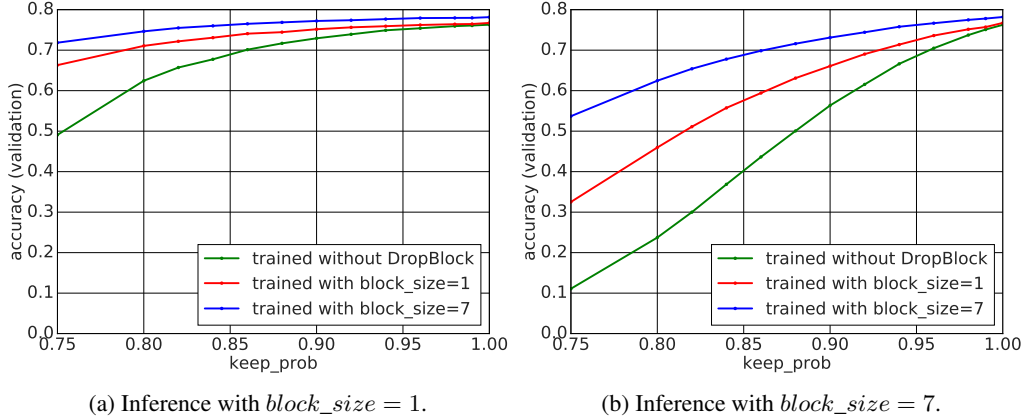

(a) Inference with $block\_size = 1$.　　　　(b) Inference with $block\_size = 7$.

Figure 5: ResNet-50 model trained with $block\_size = 7$ and $keep\_prob = 0.9$ has higher accuracy compared to the ResNet-50 model trained with $block\_size = 1$ and $keep\_prob = 0.9$: (a) when we apply DropBlock with $block\_size = 1$ at inference with different $keep\_prob$ or (b) when we apply DropBlock with $block\_size = 7$ at inference with different $keep\_prob$. The models are trained and evaluated with DropBlock at groups 3 and 4.

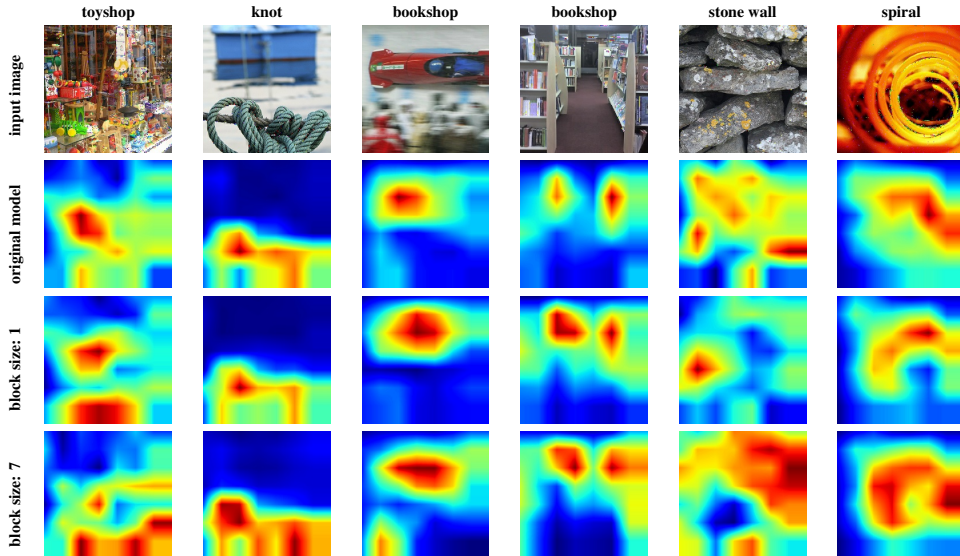

Figure 6: Class activation mapping (CAM) [29] for ResNet-50 model trained without DropBlock and trained with DropBlock with the $block\_size$ of 1 or 7. The model trained with DropBlock tends to focus on several spatially distributed regions.

more robust compared to model regularized by dropout. We study the problem by applying DropBlock with $block\_size$ of 1 and 7 during inference and observing the differences in performance.

**DropBlock drops more semantic information.** We first took the model trained without any regularization and tested it with DropBlock with $block\_size = 1$ and $block\_size = 7$. The green curves in Figure 5 show the validation accuracy reduced quickly with decreasing $keep\_prob$ during inference. This suggests DropBlock removes semantic information and makes classification more difficult. The accuracy drops more quickly with decreasing $keep\_prob$, for $block\_size = 1$ in comparison with $block\_size = 7$ which suggests DropBlock is more effective to remove semantic information than dropout.

**Model trained with DropBlock is more robust.** Next we show that model trained with large block size, which removes more semantic information, results in stronger regularization. We demonstrate

the fact by taking model trained with $block\_size = 7$ and applied $block\_size = 1$ during inference and vice versa. In Figure 5, models trained with $block\_size = 1$ and $block\_size = 7$ are both robust with $block\_size = 1$ applied during inference. However, the performance of model trained with $block\_size = 1$ reduced more quickly with decreasing $keep\_prob$ when applying $block\_size = 7$ during inference. The results suggest that $block\_size = 7$ is more robust and has the benefit of $block\_size = 1$ but not vice versa.

**DropBlock learns spatially distributed representations.**   We hypothesize model trained with DropBlock needs to learn spatially distributed representations because DropBlock is effective in removing semantic information in a contiguous region. The model regularized by DropBlock should learn multiple discriminative regions instead of only focusing on one discriminative region. We use class activation maps (CAM) introduced in [29] to visualize $conv5\_3$ class activations of ResNet-50 on ImageNet validation set. Figure 6 shows the class activations of original model and models trained with DropBlock with $block\_size = 1$ and $block\_size = 7$. In general, models trained with DropBlock learn spatially distributed representations that induce high class activations on multiple regions, whereas model without regularization tends to focus on one or few regions.

## 4.3   Object Detection in COCO

DropBlock is a generic regularization module for CNNs. In this section, we show DropBlock can also be applied for training object detector in COCO dataset [30]. We use RetinaNet [24] framework for the experiments. Unlike an image classifier that predicts single label for an image, RetinaNet runs convolutionally on multiscale Feature Pyramid Networks (FPNs) [31] to localize and classify objects in different scales and locations. We followed the model architecture and anchor definition in [24] to build FPNs and classifier/regressor branches.

**Where to apply DropBlock to RetinaNet model.**   RetinaNet model uses ResNet-FPN as its backbone model. For simplicity, we apply DropBlock to ResNet in ResNet-FPN and use the best $keep\_prob$ we found for ImageNet classification training. DropBlock is different from recent work [32] which learns to drop a structured pattern on features of region proposals.

**Training object detector from random initialization.**   Training object detector from random initialization has been considered as a challenging task. Recently, a few papers tried to address the issue using novel model architecture [33], large minibatch size [34], and better normalization layer [35]. In our experiment, we look at the problem from the model regularization perspective. We tried DropBlock with $keep\_prob = 0.9$, which is the identical hyperparamters as training image classification model, and experimented with different $block\_size$. In Table 3, we show that the model trained from random initialization surpasses ImageNet pre-trained model. Adding DropBlock gives additional 1.6% AP. The results suggest model regularization is an important ingredient to train object detector from scratch and DropBlock is an effective regularization approach for object detection.

| Model | AP | AP50 | AP75 |
|---|---|---|---|
| RetinaNet, fine-tuning from ImageNet | 36.5 | 55.0 | 39.1 |
| RetinaNet, no DropBlock | 36.8 | 54.6 | 39.4 |
| RetinaNet, $keep\_prob = 0.9$, $block\_size = 1$ | 37.9 | 56.1 | 40.6 |
| RetinaNet, $keep\_prob = 0.9$, $block\_size = 3$ | 38.3 | 56.4 | 41.2 |
| RetinaNet, $keep\_prob = 0.9$, $block\_size = 5$ | 38.4 | 56.4 | 41.2 |
| RetinaNet, $keep\_prob = 0.9$, $block\_size = 7$ | 38.2 | 56.0 | 40.9 |

Table 3: Object detection results trained from random initialization in COCO using RetinaNet and ResNet-50 FPN backbone model.

**Implementation details.**   We use open-source implementation of RetinaNet[3] for experiments. The models were trained on TPU with 64 images in a batch. During training, multiscale jitter was applied to resize images between scales [512, 768] and then the images were padded or cropped to max dimension 640. Only single scale image with max dimension 640 was used during testing. The batch

normalization layers were applied after all convolution layers, including classifier/regressor branches. The model was trained using 150 epochs (280k training steps). The initial learning rate 0.08 was applied for first 120 epochs and decayed 0.1 at 120 and 140 epochs. The model with ImageNet initialization was trained for 28 epochs with learning decay at 16 and 22 epochs. We used $\alpha = 0.25$ and $\gamma = 1.5$ for focal loss. We used a weight decay of 0.0001 and a momentum of 0.9. The model was trained on COCO train2017 and evaluated on COCO val2017.

### 4.4 Semantic Segmentation in PASCAL VOC

We show DropBlock also improves semantic segmentation model. We use PASCAL VOC 2012 dataset for experiments and follow the common practice to train with augmented 10,582 training images [36] and report mIOU on 1,449 test set images. We adopt open-source RetinaNet implementation for semantic segmentation. The implementation uses the ResNet-FPN backbone model to extract multiscale features and attaches fully convolution networks on top to predict segmentation. We use default hyperparameters in open-source code for training.

Following the experiments for object detection, we study the effect of DropBlock for training model from random initialization. We trained model started with pre-trained ImageNet model for 45 epochs and model with random initialization for 500 epochs. We experimented with applying DropBlock to ResNet-FPN backbone model and fully convolution networks and found apply DropBlock to fully convolution networks is more effective. Applying DropBlock greatly improves mIOU for training model from scratch and shrinks performance gap between training from ImageNet pre-trained model and randomly initialized model.

| Model | mIOU |
|---|---|
| fine-tuning from ImageNet | 74.6 |
| no DropBlock | 47.2 |
| $keep\_prob = 0.2, block\_size = 1$ | 51.0 |
| $keep\_prob = 0.2, block\_size = 4$ | 53.2 |
| $keep\_prob = 0.2, block\_size = 16$ | 53.4 |

Table 4: Semantic segmentation results trained from random initialization in PASCAL VOC 2012 using ResNet-101 FPN backbone model.

## 5 Discussion

In this work, we introduce DropBlock to regularize training CNNs. DropBlock is a form of structured dropout that drops spatially correlated information. We demonstrate DropBlock is a more effective regularizer compared to dropout in ImageNet classification and COCO detection. DropBlock consistently outperforms dropout in an extensive experiment setup. We conduct an analysis to show that model trained with DropBlock is more robust and has the benefits of model trained with dropout. The class activation mapping suggests the model can learn more spatially distributed representations regularized by DropBlock.

Our experiments show that applying DropBlock in skip connections in addition to the convolution layers increases the accuracy. Also, gradually increasing number of dropped units during training leads to better accuracy and more robust to hyperparameter choices.

## Footnotes

[1]https://github.com/tensorflow/tpu/tree/master/models/official/resnet

[2]https://github.com/tensorflow/tpu/tree/master/models/experimental/amoeba_net

[3]https://github.com/tensorflow/tpu/tree/master/models/official/retinanet

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
