[Reviews · NeurIPS 2018]

Reviewer 1



This work proposes a regularization method similar to dropout but specifically designed for convolutional blocks. Rather than dropping random neurons, and therefore disregarding the local correlation in the feature maps, the authors propose to drop contiguous blocks of neurons. This way, some of the semantic information in the features can be dropped altogether. This new regularization method is evaluated on state-of-the-art architectures for image classification and object detection, and shows improvement of the accuracy when compared to dropout and other similar methods. I recommend acceptance of the paper for the following reasons: Originality: To my knowledge, the idea is novel, and well motivated. Clarity: The paper is clear and well structured, both for the method description, and for the experiments. The experiments are also well detailed, and reproduction of the work should be possible. Quality: The authors provided a thorough explanation of each of the parameters of the method both in the description of the proposed regularization and in the conducted experiments. They also considered multiple schedules for the parameters of the method: fixed and increasing drop probability. Moreover, they compared the proposed method to multiple related works. Significance: Although the shown results are already interesting, it would be interesting to see how it works on segmentation models.

Reviewer 2



Summary: This paper proposes a variant version of dropout called DropBlock as training regularization for deep neural networks. Rather than drop features randomly, DropBlock put a spatial mask on each dropout center so that a spatial region is dropped out in training. It claims this kind of dropout can better drop more semantic information. The experiments show that on ResNet50 the dropblock brings about 0.5% - 1% improvement over the baselines. Strengths: + A variant of dropout is proposed to improve the training of deep neural networks. It is a general training regularizer which could be plug in on a lot of different architectures. + It shows improvement over resnet for imagenet classification and retinaNet for object detection. + The sensitivity analysis of the parameters has been done. Weakness: - One drawback is that the idea of dropping a spatial region in training is not new. Cutout [22] and [a] have been explored this direction. The difference towards previous dropout variants is marginal. [a] CVPR'17. A-Fast-RCNN: Hard Positive Generation via Adversary for Object Detection. - The improvement over previous methods is small, about 0.2%-1%. Also the results in Table 1 and Fig.5 don't report the mean and standard deviation, and whether the difference is statistically significant is hard to know. I will suggest to repeat the experiments and conduct statistical significance analysis on the numbers. Thus, due to the limited novelty and marginal improvement, I suggest to reject the paper.

Reviewer 3



The authors present a version of dropout that zeros out square blocks of neurons rather than individual neurons. They experience robust improvements relative to dropout. This is a good paper, with potentially very wide-spread applicability. The authors only present validation error in all their studies, but not test error. They claim that this is "common practice". I quickly checked the ResNet / DenseNet / Stochastic Depth / ShakeShake and ShakeDrop papers. Only one paper (DenseNet) seems to use validation error consistently and one (ResNet) partially. So I wouldn't call reporting the validation error common practice. In any case, this is a terrible practice for obvious reasons so I would urge any author (including the authors of this paper) to break the trend by reporting test error. If the authors feel that reporting validation error is necessary for comparion, they can report both. Did you use the validation set for hyperparameter tuning? If so, in what way? Since you write "we first estimate the block_size_factor to use (typically 0.5)", it seems you are using it for at least some tuning. How is reporting validation error on the final model then still unbiased? I am confused by your treatment of the DropBlock hyperparameters. On page 3, you start by saying that you don't set gamma and block_size directly, but block_size_factor and keep_prob. But then in line 156 you say "gamma = 0.4". I am assuming you mean "keep_prob = 0.4"? Similarly, in table 2 / line 210, you use gamma, but I think you mean keep_prob. How do you compensate for DropBlock? E.g. in dropout, if a feature is dropped with probability gamma, all other outputs are multiplied with 1/(1-gamma). How do you handle this in DropBlock? Have you tried block_size_factor=1, i.e. dropping entire channels? What happens then? ### Additional comments post-rebuttal ### I apologize for the stupid comment regarding the imagenet test set. I didn't know that it is "secret". "We are happy to create a holdout partition of training data and re-do the experiments for the final submission of 33 the paper." I am envious of your seemingly unlimited GPU access :) But seriously, I wouldn't want to burden you with this is everyone else if using the validation set for final numbers. I will leave it up to you whether you think having a test partition would improve the paper. The block_size_factor=1 results are interesting. Maybe you can find space to include them in the final paper. Also, you write "we first estimate the block_size_factor to use (typically 0.5)". I'd like to see more details on how this estimation happened. Also, I can't find drop_block_size for figure 4 (it's probably 0.5 ...). In figure 4, right side, "after sum" is missing an 'm'.